# From Transparent to Opaque: Rethinking Neural Implicit Surfaces with $\alpha$-NeuS

**Haoran Zhang**[1,2]*, **Junkai Deng**[3]*, **Xuhui Chen**[1,2]*, **Fei Hou**[1,2]†, **Wencheng Wang**[1,2],
**Hong Qin**[4], **Chen Qian**[5], **Ying He**[3]

[1]Key Laboratory of System Software (CAS) and State Key Laboratory of Computer Science,
Institute of Software, Chinese Academy of Sciences
[2]University of Chinese Academy of Sciences
[3]College of Computing and Data Science, Nanyang Technological University
[4]Department of Computer Science, Stony Brook University      [5]SenseTime Research
`zhanghr@ios.ac.cn` `junkai006@e.ntu.edu.sg` `{chenxh, houfei, whn}@ios.ac.cn`
`qin@cs.stonybrook.edu` `qianchen@sensetime.com` `yhe@ntu.edu.sg`

## Abstract

Traditional 3D shape reconstruction techniques from multi-view images, such as structure from motion and multi-view stereo, face challenges in reconstructing transparent objects. Recent advances in neural radiance fields and its variants primarily address opaque or transparent objects, encountering difficulties to reconstruct both transparent and opaque objects simultaneously. This paper introduces $\alpha$-NeuS—an extension of NeuS—that proves NeuS is unbiased for materials from fully transparent to fully opaque. We find that transparent and opaque surfaces align with the non-negative local minima and the zero iso-surface, respectively, in the learned distance field of NeuS. Traditional iso-surfacing extraction algorithms, such as marching cubes, which rely on fixed iso-values, are ill-suited for such data. We develop a method to extract the transparent and opaque surface simultaneously based on DCUDF. To validate our approach, we construct a benchmark that includes both real-world and synthetic scenes, demonstrating its practical utility and effectiveness. Our data and code are publicly available at `https://github.com/728388808/alpha-NeuS`.

## 1 Introduction

Surface reconstruction from multi-view images has been an important area of research for decades. Traditional methods such as structure from motion (SfM) [1] and multi-view stereo (MVS) [2] calibrate images and reconstruct 3D geometry based on color consistency. Recently, the emergence of Neural Radiance Fields (NeRF) [3] has revolutionized the field, producing impressive results in novel view synthesis results via volume rendering. Its implicit surface-based variants, such as NeuS [4], VolSDF [5], HF-NeuS [6], and NeuS2 [7], further advance this field by reconstructing high-quality geometry and appearance through the learning of signed distance fields (SDFs). However, these NeRF family methods are limited to reconstructing opaque surfaces. Reconstructing transparent surfaces presents a greater challenge, with relatively few investigations to date.

Recently, some works dealt with the refraction and reflection effects in the transparent scenes. For example, ReNeuS [8] effectively reconstructs opaque objects within transparent materials, such as glass, by assuming known parameters for these materials. Similarly, NeuS-HSR [9] separates

---

*Equal contributions.
†Corresponding author.

38th Conference on Neural Information Processing Systems (NeurIPS 2024).

reflections from the glass to reconstruct objects within thin transparent objects. While these methods successfully reconstruct the opaque objects behind or within transparent materials, they do not extend to reconstructing the transparent objects themselves.

To address the challenges described above, we propose a new method, called $\alpha$-NeuS, for the simultaneous reconstruction of thin transparent objects and opaque objects. Given that transparent objects are thin, we can disregard refraction effects. A key observation in our work is that transparent surfaces induce local extreme values in the learned distance field of NeuS [4] during neural volumetric rendering. NeuS is unbiased, i.e., the maximum volume rendering weight coincides with the object surface, for opaque surface [4]. We advance the theory of NeuS and prove that NeuS is unbiased for all transparent and opaque surfaces. Under various opacities, the unbiased surfaces are either the non-negative local minimum or the zero level set of the distance field learned by NeuS. Thus, we are able to extract the unbiased surface for transparent and opaque surface reconstruction simultaneously. However, precise values of these non-negative local minima are unknown beforehand and can vary spatially, they are unsuitable for extraction by conventional iso-surfacing algorithms, such as marching cubes [10], which require a specified fixed iso-value. To effectively extract the target geometry for transparent objects, we take the absolute value of the distance fields, making the unbiased surfaces become the local minima of the absolute distance field. Based on DCUDF [11], we introduce an optimization method to simultaneously extract the unbiased surfaces of the transparent and opaque surfaces.

To validate our approach, we construct a benchmark containing 5 real-world scenes and 5 synthesized scenes. Experimental results show that $\alpha$-NeuS effectively reconstructs both transparent and opaque objects in all tested scenarios. To summarize, our main contributions are as follows:

1. We prove that the density functions proposed in NeuS [4] are unbiased across a continuum of material opacities, from fully transparent to fully opaque, thereby completing the theoretical framework of NeuS.

2. We show that transparent and opaque surfaces correspond to the non-negative local minima and the zeros of the learned distance field of NeuS, respectively.

3. We present a method for simultaneously extracting the unbiased surfaces corresponding to the target geometry of transparent objects and opaque objects based on DCUDF [11], from mixed SDF and unsigned distance field (UDF).

4. We construct a benchmark comprised of 5 real-world scenes and 5 synthetic scenes for validating our method.

## 2 Related Works

### 2.1 3D reconstruction from multi-view images

Reconstructing 3D objects from multi-view 2D images has been a research interest for decades, with a wide range of approaches having been proposed. Traditional model structure recovery methods try to understand the images and infer the structure of the model. Notable examples in this category are voxel based approaches [12–16] and point cloud based approaches including SfM [1] and MVS [2].

Recently, with the advancement of machine learning, volume rendering based approaches have achieved high-fidelity reconstruction quality. Based on 3D Gaussian splatting [17], many model reconstruction methods are proposed, e.g., SuGaR [18] and 2D Gaussian splatting [19]. Another category of method for surface reconstruction is NeuS [4] and VolSDF [5], based on NeRF [3]. In particular, NeuS has gathered special attention and has spawned multiple descendants like Geo-Neus [20] and HF-NeuS [6]. There are also several studies that focus on non-watertight model reconstruction also by extending NeuS, including NeUDF [21], NeuralUDF [22], NeAT [23] and 2S-UDF [24]. However, these works all assume that the object is opaque.

### 2.2 3D reconstruction of transparent objects

Reconstruction of transparent objects presents a significant challenge due to the complex light paths [30] caused by refraction and reflection, which hinder multiview stereo from effectively solving this problem [31]. Traditional methods [32–34] use additional devices or assumptions to

Table 1: Comparison with related works for respective use cases.

| Method | Opaque | Transparent | Refraction | Reflection | Note |
|--------|--------|-------------|------------|------------|------|
| NeuS [4] | Yes | No | No | No | |
| ReNeuS [8] | Yes | No | Yes | Yes | 1,3 |
| NeuS-HSR [9] | Yes | No | No | Yes | 1 |
| TransPIR [25] | N/A | Yes | Yes | Yes | 2,5 |
| Li. 2020 [26] | N/A | Yes | Yes | Yes | 2,4,5 |
| NeTO [27] | N/A | Yes | Yes | No | 2 |
| RefNeuS [28] | Yes | No | No | Yes | |
| $\alpha$Surf [29] | Yes | Yes | No | No | |
| $\alpha$-NeuS (Ours) | Yes | Yes | No | No | |

**Notes:**
[1]Opaque object inside transparent container.
[2]Focuses on pure glass objects.
[3]Assumes known container geometry and homogeneous background lighting.
[4]Assumes maximum two light bounces.
[5]Not involving volume rendering.

reconstruct transparent objects. Li et al. [26] used deep learning to further improve the quality of reconstruction without additional inputs, but tend to produce over-smoothing results. With the development of NeRF [3], some works have explored how to use neural rendering to reconstruct transparent objects [25, 27, 35–37] for capturing more details. However, they all just focus on transparent objects neglecting opaque objects.

There are also some works attempting to capture the correct geometry of opaque objects under the influence of reflection and refraction, which are primarily caused by transparent objects. NeRFReN [38] and Ref-NeuS [28] reconstruct models by considering reflections in NeRF pipeline. NeuS-HSR [9] uses a similar idea to model opaque objects inside transparent objects by separating the reflection effect. ReNeuS [8] considers both reflection and refraction to model the opaque object inside glass, but needs strong assumption. These methods focus on the reconstruction of opaque models while overcoming the interference of reflection and refraction.

But none of them can reconstruct both transparent and opaque objects. They all have their own assumptions or conditions for reducing the effect of reflection or refraction. It is non-trivial to combine the two tasks. Please refer to Table 1 for a comprehensive comparison on the use cases. A concurrent work [39] proposed a NeRF-based efficient rendering method for non-opaque scenes with baked quadrature fields. Another concurrent work, $\alpha$Surf [29], extends Plenoxels [40] for modeling both transparent objects and opaque objects, while ignoring the effect of reflection and refraction. Concurrently, the work NU-NeRF [41] tries to model both transparent and opaque objects while recovering refractions. In this paper, we propose a new algorithm to reconstruct thin transparent objects and opaque objects uniformly based on NeuS [4].

## 3 Method

### 3.1 Unbiased density mapping in NeuS across opacities

NeuS [4] utilizes signed distance fields for surface representation and introduces a density distribution induced by these SDFs, thereby enabling neural volume rendering coupled with SDF learning. NeuS [4] proved that for opaque objects, the mapping from SDF to density in NeuS is unbiased, ensuring that the reconstructed surface is a first-order approximation of the learned SDF. In this section, we further establish that the density mapping proposed in NeuS is indeed unbiased across a continuum of material opacities, from fully transparent to fully opaque. This verification completes the theoretical framework of NeuS.

A surface is considered unbiased if the rendering weights attain the local maxima on the surface. This is essential to minimize the discrepancy between the surface and the desired result. NeuS [4] assumes the surface is opaque and proved the zero iso-surface is unbiased. For transparent surface,

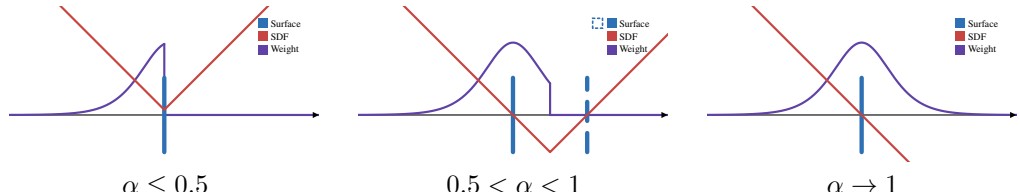

$$\alpha \leq 0.5 \qquad\qquad 0.5 < \alpha < 1 \qquad\qquad \alpha \to 1$$

Figure 1: Conceptual illustration of the signed distances along a ray (black horizontal line) through a scene containing a single object (represented by a vertical line segment). (a) When the rendered opacity $\alpha$ is less than or equal to $0.5$, both the front and back faces of the object coincide with each other, aligning the maximal weight with the local minimum of the distance field. (b) When $\alpha$ exceeds $0.5$ but less than $1$, the back face, which is not rendered, is separated from the front face. The further away the back face is, the more opaque the rendered front face is. The maximum weight in this case is aligned with the position of zero distance values. (c) For a fully opaque surface, the back face is infinitely away. The scene can therefore be considered the single ray-plane intersection discussed by NeuS [4].

we observed that NeuS can also produce a local minimum distance on the transparent surface, which inspired us to explore the properties of these local minima. We have the following theorem.

**Theorem 1.** *Assuming a single ray-plane intersection[3], if the rendered opacity $\alpha \leq 0.5$, the learned distance field reaches a local minimum which is non-negative, and the corresponding color weight maximum aligns with the distance local minimum. Otherwise, the distance local minimum is smaller than zero, and the corresponding color weight maximum aligns with the zero iso-surface.*

Please refer to Appendix A for the details of the proof. We simply sketch the proof here. Assume $\alpha$ is the opacity and the ray starting from point $\mathbf{o}$ in direction $\mathbf{d}$ is $\mathbf{r}(t) = \mathbf{o} + t\mathbf{d}$ with parameter $t$. The density function of NeuS [4] and HF-NeuS [6] is

$$\sigma(\mathbf{r}(t)) = \max\left(-s\left(1 - \Phi_s(f(\mathbf{r}(t)))\right)\cos(\theta), 0\right), \tag{1}$$

where $s$ is a learnable parameter, $\Phi_s(\cdot)$ is the sigmoid function and $\theta$ is the angle between the ray direction and the gradient of the distance field $f$. The $\max$ operation avoids negative $\sigma$ after crossing the local minimum $m$ of the distance field. Assume the distance between origin of the ray and the plane is $d_0$. Then, the opacity is

$$\alpha = \begin{cases} \frac{1 - e^{-sd_0}}{1 + e^{sm}}, & m \geq 0 \\ 1 - \frac{1 + e^{-sd_0}}{1 + e^{-sm}}, & m < 0 \end{cases} \tag{2}$$

If $m = 0$, the opacity $\alpha = \frac{1 - e^{-sd_0}}{2} \approx 0.5$, since $s$ and $d_0$ are relatively large. For the sake of brevity, we simply state $0.5$ as the watershed value in the theorem.

The derivative of the rendering weight $w(t)$ is $w'(t) = T(t)\left[\sigma'(\mathbf{r}(t)) - \sigma^2(\mathbf{r}(t))\right]$. If the local maximum of weight occurs at $t = t^*$, $\sigma'(\mathbf{r}(t^*)) = \sigma^2(\mathbf{r}(t^*))$. Taking the density function of NeuS into the above equation, we have $f(\mathbf{r}(t^*)) = 0$, if $m < 0$. Thus, if $m < 0$, the zero iso-surface attains the largest rendering weight. We can also deduce that $w'(t) > 0$ if $m > 0$, and thus the rendering weight continuously increases until touching the minimum of the distance field. Therefore, the local minimum of the distance field attains the largest rendering weight. In case of $m = 0$, the point where the distance reaches minimum also achieves zero distance value, so the rendering weight is maximum at the distance local minimum.

The theorem can be explained as follows. As illustrated in Figure 1, if $\alpha \leq 0.5$, the distance field local minimum is non-negative and the unbiased surface coincides with the local minimum. If $0.5 < \alpha < 1$, the local minimum is negative and the unbiased surface coincides with the front zero iso-surface. If $\alpha \to 1$, the local minimum approaching negative infinity and the back zero iso-surface approaching infinity. Along with the opacity $\alpha$ increasing, the front and back faces separate gradually from overlap to infinity, so that the integral of densities increases to infinity gradually.

---

[3]With this assumption, we focus on first-order unbiasedness.

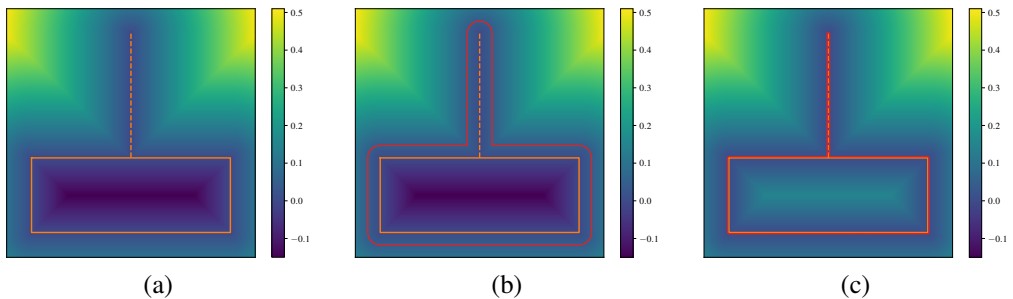

(a)  (b)  (c)

Figure 2: Illustration of our mesh extraction procedure. (a) The orange line denotes the input model, where the dashed line is transparent and the solid line is opaque. The color map illustrates the distance field $f$. (b) The $r$ iso-curve (red) is extracted. (c) The iso-curve is mapped to the local minima of the absolute distance field $f^a$.

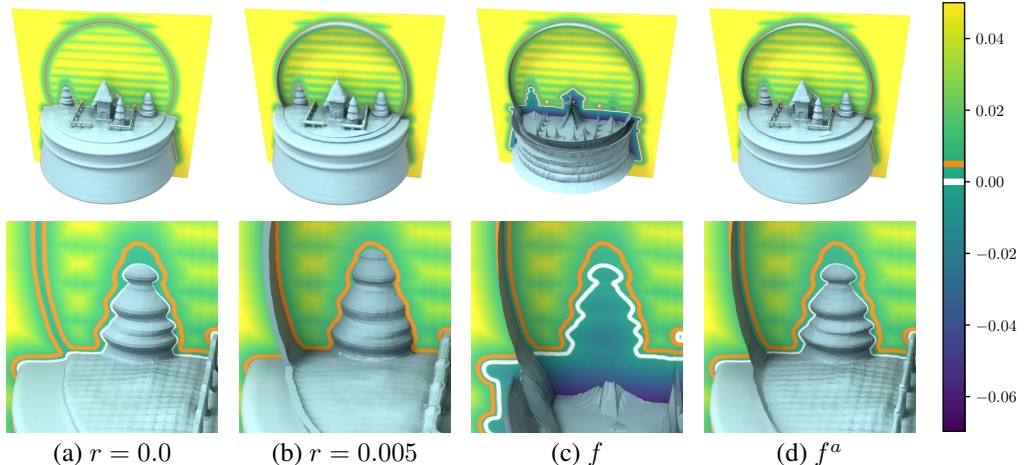

(a) $r = 0.0$     (b) $r = 0.005$     (c) $f$     (d) $f^a$

Figure 3: Comparisons of projection on the mixed SDF and UDF $f$ and the absolute field $f^a$. The cutting plane draws the distance field. The white line indicates the 0 iso-surface and the orange line indicates the 0.005 iso-surface. (a) The extracted 0 iso-surface which attains the opaque surface exactly, but the transparent surface disappears. (b) The extracted 0.005 iso-surfaces. (c) Direct mapping on the original $f$ would result in the opaque surfaces shrinking. (d) In contrast, after taking the absolute, all unbiased surfaces are properly extracted.

*Remark.* In NeuS, $s$ is a learnable parameter that gradually converges to a large value over the course of training. A larger $s$ value sharpens the edges and faces in the reconstructed model, enhancing overall quality. However, in practice, $s$ cannot increase indefinitely due to numerical computation constraints, such as the number of sample points. This caps $s$ at relatively high but finite values, which allows non-zero distance values to influence color calculation along the ray. Colors are derived from the the weighted sum of sampled radiance at points along the rays, and different distance minimum values contributes to achieving different levels of opacity.

We believe that the tendency of a larger $s$, combined with the color loss and the actual situations with MLP and numerical calculation will achieve a balance that leads to the best results.

## 3.2 Unbiased surface extraction

As mentioned in Section 3.1, we aim to extract the unbiased surface from the learned distance field. The unbiased surface is either the local minimum or the zero iso-surface depending on whether the local minimum is non-negative. Thus, the distance field learned by NeuS is not a UDF or an SDF. As illustrated in Figure 1, when $\alpha \leq 0.5$, the distance field is similar to a UDF whose values are positive on both side and the unbiased surface is the local minimum. Since the local minimum is greater than or equal to zero, the distance field is not a strict UDF. However, for simplicity, we still call such

distance field a UDF. When $\alpha > 0.5$, the distance field is an SDF whose values are positive in front of the surface and negative behind the surface. The unbiased surface is the zero iso-surface and the zero is not an extreme value. Hence, the unbiased surface is a mixed SDF and UDF, which cannot be extracted using the conventional iso-surface extraction methods, e.g., marching cubes [10]. In Figure 3(a), the zero iso-surface cannot extract the transparent hemisphere.

To extract the unbiased surface from the mixed SDF and UDF, we follow the idea of DCUDF [11] to extract the unbiased surface. As illustrated in Figure 2(a), given the mixed distance field $f$ learned by NeuS, we extract the mesh $\mathcal{M}$ of a non zero level set with a user-specific iso-value $r$ $(r > m)$ by marching cubes [10] (Figure 2(b) and 3(b)). $\mathcal{M}$ encloses the intended unbiased surface as an envelop. As DCUDF [11], we compute a covering map to project $\mathcal{M}$ back to the local minima. However, if $\alpha > 0.5$, the unbiased surfaces are not the local minima, but the zero iso-surface. In Figure 2(b), the values of $f$ inside the opaque box are smaller than the values outside. If we project $\mathcal{M}$ to the local minima of $f$, the curve would shrink into the box. Figure 3(c) shows an example that the opaque surface shrinks if we project $\mathcal{M}$ to the local minima of $f$. Nevertheless, as shown in Figure 2(c) and 3(d), if we convert $f$ into its absolute values denoted by $f^a$, the non-negative local minima and the zero iso-surface of $f$ are both the local minima of $f^a$. Thus, $f^a$ is a UDF whose local minima are the unbiased surfaces. We are able to map $\mathcal{M}$ to the local minima to extract the unbiased surfaces. Following DCUDF [11], which employs a two-stage optimization process, we first solve a mapping $\pi_1$ to project $\mathcal{M}$ to the local minima of $f^a$ [11]:

$$\min_{\pi_1} \sum_{p_i \in \mathcal{M} \cup \mathcal{C}} f^a\big(\pi_1(p_i)\big) + \lambda_1 \sum_{p_i \in \mathcal{M}} w(p_i)\big\|\Delta\pi_1(p_i)\big\|^2, \text{where}$$

$$\Delta\pi_1(p_i) = \pi_1(p_i) - \frac{1}{|\mathcal{N}(p_i)|} \sum_{p_j \in \mathcal{N}(p_i)} \pi_1(p_j)$$

is the Laplacian of the projected point $\pi_1(p_i)$ and $\mathcal{C}$ is the set of triangle centroids of $\mathcal{M}$. $f(\pi_1(p_i))$ drives the point $p_i$ projecting to the local minima of $f^a$. $\mathcal{N}(p_i)$ denotes the 1-ring neighboring vertices of $p_i \in \mathcal{M}$ and $w(p_i)$ is a weight adaptive to the area of the adjacent triangle faces of $p_i$. The second term is a Laplacian constraint that prevents the mesh from folding and self-intersecting during optimization.

DCUDF [11] further calculates a mapping $\pi_2$ to refine $\pi_1(\mathcal{M})$ in the stage two, which further reduces the fitting error. $\overrightarrow{n}_i$ denotes the normal of the $i$-th triangle face of $\pi_1(\mathcal{M})$, whose centroid is encouraged to move along the normal direction $\overrightarrow{n}_i$ by penalizing the tangential displacements so as to prevent mesh folding and self-intersecting. The loss function of the refinement stage is [11]:

$$\min_{\pi_2} \sum_{p_i \in \mathcal{M} \cup \mathcal{C}} f^a\big(\pi_2 \circ \pi_1(p_i)\big) + \lambda_2 \sum_{p_i \in \mathcal{C}} \big\|\big(\pi_2 \circ \pi_1(p_i) - \pi_1(p_i)\big) \times \overrightarrow{n}_i\big\|,$$

where $\times$ is the vector cross product. After projection, the initial $\mathcal{M}$ shrinks to the unbiased surfaces as illustrated in Figure 3(d). Since the surface may contain non-manifold structures, e.g., the intersection of the transparent and opaque surfaces in Figure 2, we do not apply the min-cut postprocessing as [11]. Hence, the unbiased surface is a two-layer mesh that coincides together in regions of $m \geq 0$ (i.e., $\alpha \leq 0.5$), and a single-layer mesh in regions of $m < 0$ (i.e., $\alpha > 0.5$).

## 4 Experiments

### 4.1 Experimental settings

**Datasets.** Due to the absence of relevant datasets, we have prepared a dataset comprised of 5 synthetic scenes and 5 real-world scenes. The synthetic data are rendered using Blender. The real data are captured by ourselves and the camera are calibrated with the help of ArUco calibration boards.

**Baselines.** We compare our method with the original NeuS [4], and NeUDF [21] which learns UDF from multi-view images.

**Implementation details.** Our training structure is the same as NeuS. We also followed the recommended configuration for the synthetic dataset by the authors of NeuS, without changing the loss

Table 2: Quantitative evaluation ($\times 10^{-3}$) on the synthetic dataset. "g2d" is the Chamfer distance from the ground truth mesh to the reconstructed mesh, and "d2g" measures the reverse. "CD" denotes the average of "g2d" and "d2g". The best results are marked in bold.

| | NeuS [4] (iso = 0) | | | NeuS [4] (iso = 0.005) | | | Ours | | |
|---|---|---|---|---|---|---|---|---|---|
| | g2d | d2g | CD | g2d | d2g | CD | g2d | d2g | CD |
| Snowglobe | 65.22 | 5.16 | 35.19 | 7.07 | 6.46 | 6.77 | **4.73** | **4.37** | **4.55** |
| Case | 39.60 | 8.18 | 23.89 | 6.23 | 8.80 | 7.51 | **5.52** | **7.66** | **6.59** |
| Bottle | 7.91 | 4.77 | 6.34 | 6.19 | 8.14 | 7.16 | **3.14** | **4.22** | **3.68** |
| Jug | 11.59 | 10.41 | 11.00 | 5.33 | 9.36 | 7.34 | **2.89** | **6.44** | **4.67** |
| Jar | 76.79 | **4.45** | 40.62 | **11.61** | 7.92 | 9.77 | 11.89 | 5.87 | **8.88** |
| mean | 40.22 | 6.60 | 23.41 | 7.29 | 8.14 | 7.71 | **5.63** | **5.71** | **5.67** |

functions or their respective weights. That is, we chose $\lambda_1 = 1.0$ for color loss and $\lambda_2 = 0.1$ for Eikonal loss. All our trainings are without mask.

To extract the unbiased surface through DCUDF [11], we choose to use 0.005 as the threshold for synthetic scenes, and 0.002 or 0.005 for real-world scenes. We conducted our experiments using almost the same setting as DCUDF. DCUDF employs a two-stage optimization process. We performed 300 epochs for step 1 and 100 epochs for step 2 respectively, which is the default setting of DCUDF. We used the VectorAdam optimizer as suggested by DCUDF. We set the weights $\lambda_1 = 500$ and $\lambda_2 = 0.5$, which are different from DCUDF default setting.

The training process of NeuS typically takes about 9.75 hours and DCUDF convergence only requires a few minutes on a single NVIDIA A100 GPU.

## 4.2 Synthetic data

In this section we focus on the Synthetic Blender dataset, where each synthetic dataset comprises 100 training images from different viewpoints. We compared the reconstruction results with NeuS and compared the Chamfer Distance (CD) with NeuS on the threshold 0 and 0.005. We report the Chamfer distance results in Table 2. The results indicate that our approach can effectively reconstruct unbiased surfaces, both transparent and opaque. Please refer to Figure 4 for a qualitative comparison.

In comparison to the vanilla NeuS [4] from which we extract zero iso-surface, large portions of transparent surfaces are absent from the reconstructed mesh due to the positive minimum distances. Consequently, the one-way Chamfer distance from the ground truth to the reconstructed mesh is considerably high. Figure 5 illustrates the percentage of sample points on the ground truth models, whose distances are smaller than the threshold. The final rates reflect the completeness of the reconstructed models. It is evident that our method all achieves 100% completeness, indicating the absence of unnecessary holes in our models. In contrast, if extracting the zero iso-surface, there are approximately 20% to 40% holes remaining. In comparison to the vanilla NeuS from which we extract 0.005 iso-surface, the extracted surface does not correspond to the maximum color weight, leading to sub-optimal reconstructed-to-ground-truth one-way Chamfer distance. Our method preserves transparent surfaces while being unbiased, leading to the best Chamfer distances across all data.

## 4.3 Real-world data

We also capture 5 real-world scenes for validation. Due to the absence of ground-truth mesh data, qualitative comparisons are conducted with NeuS on real-world scenes. The results are presented in Figure 6. The visual results demonstrate that our method exhibits good reconstruction quality for both transparent and opaque surfaces, even in complex lighting conditions in real-world scenes. In contrast, NeuS with zero iso-surfaces is unable to extract a completed surface, resulting in artifacts.

## 4.4 Discussion

**Choice of NeuS.** We use NeuS [4] as backbone for reconstruction, which learns a mixed SDF and UDF. However, during the projection stage of surface extraction, we use the absolute value

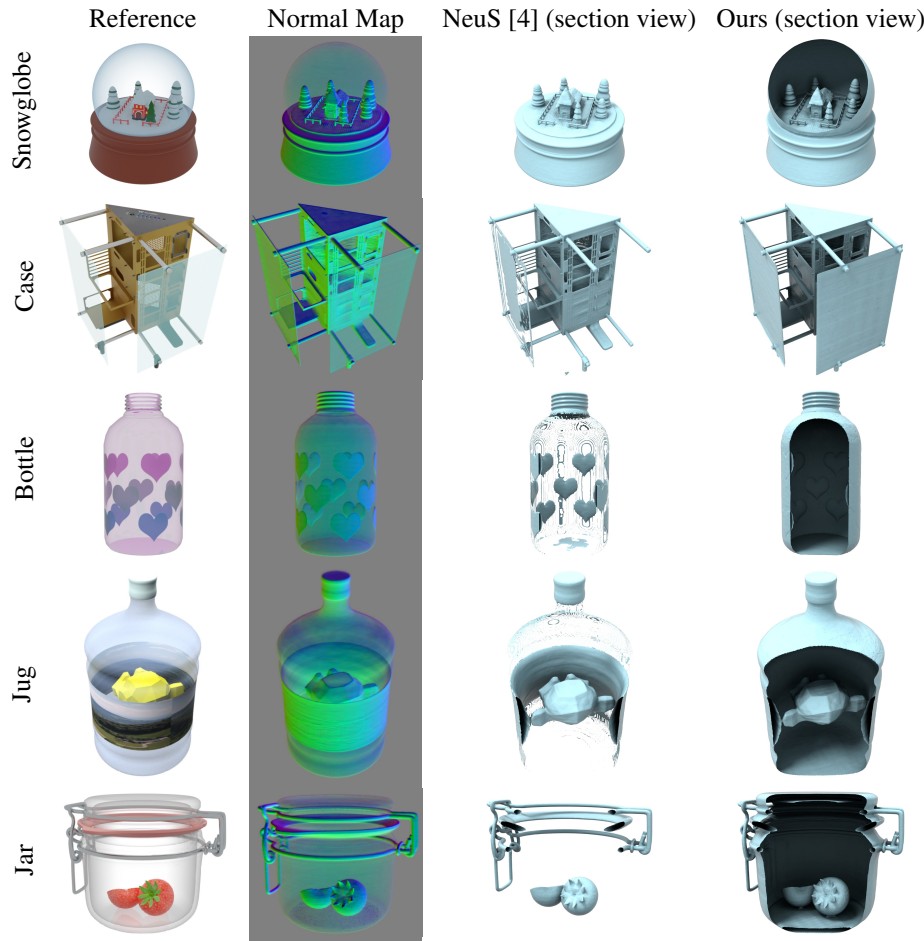

Figure 4: Qualitative comparison on synthetic data. Our method uses NeuS for distance field learning, and as shown in the normal maps, vanilla NeuS is in fact capable of reconstructing surfaces with transparency. The difference between our method and NeuS is drastic because NeuS cannot extract transparent surfaces where the distance field local minima are larger than zero with marching cubes, but our theory confirms and extends NeuS's learning ability, extracting both the non-negative local minima and the zero iso-surface.

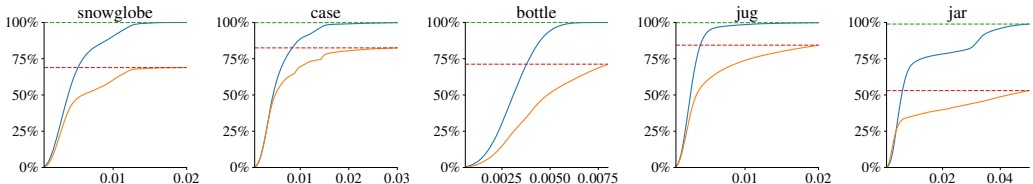

Figure 5: Percentage of sample points on the ground truth mesh that the distance to the reconstructed mesh is lower than given values. Blue: Ours, Orange: Zero iso-surface.

of the distance field. The absolute distance field $f^a$ resembles UDF. While directly using UDF learning methods could avoid the distance field conversion process, we select NeuS rather than UDF learning methods because the the SDF learning method NeuS is simple, stable and robust, and is also capable of reconstructing details. We further compare with a UDF learning method NeUDF [21]. We notice that other UDF learning methods including NeuralUDF [22] and 2S-UDF [24] both take advantage of the opaque surface assumption, introducing an indicator function or ray truncation

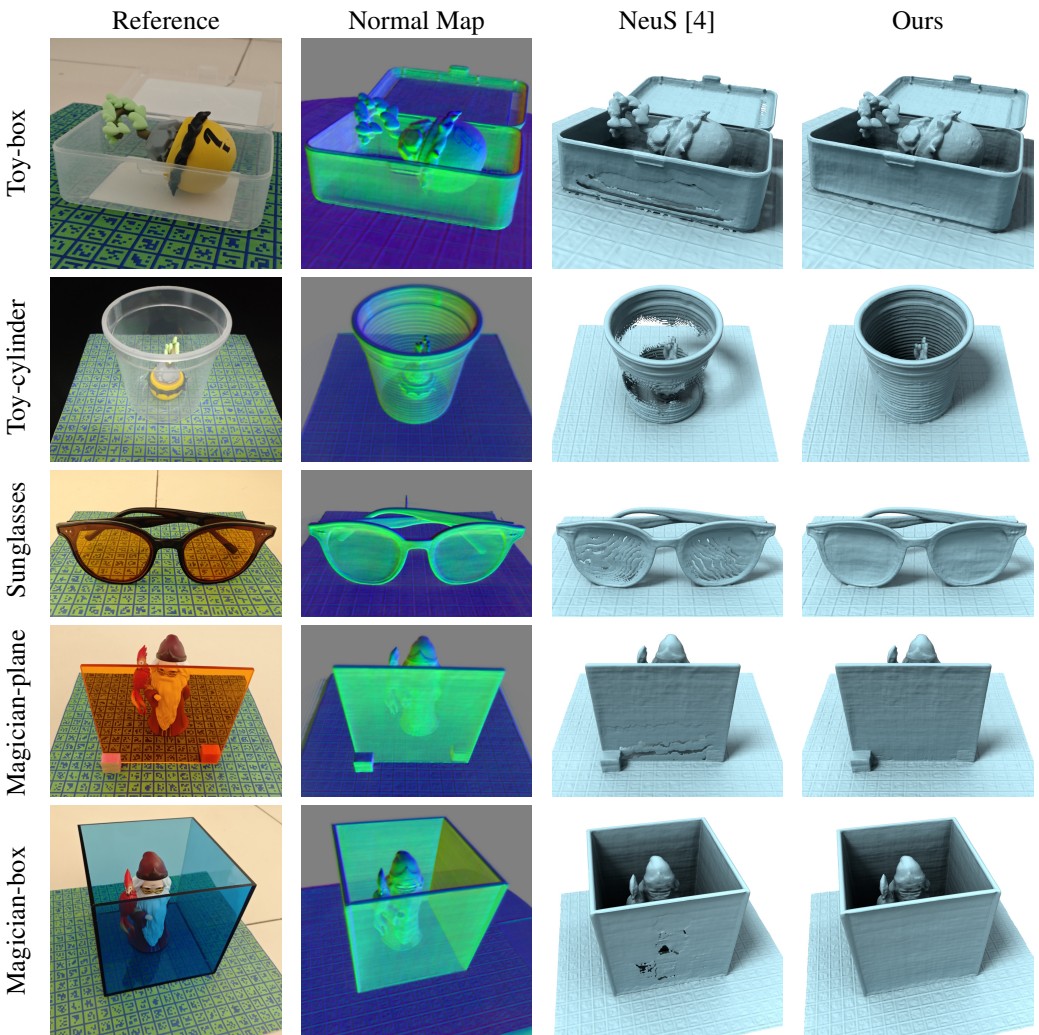

Figure 6: Qualitative comparisons on real-world data.

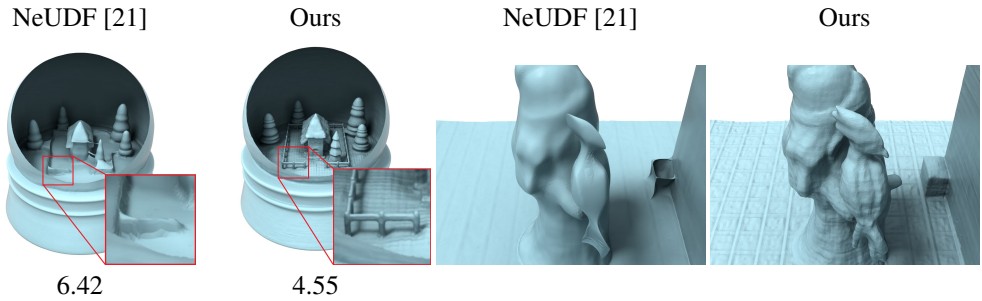

Figure 7: Comparisons with UDF-based reconstruction method NeUDF [21] on synthetic and real-world data. While NeUDF can successfully reconstruct transparent surfaces and interior structures, it fails to preserve details and has difficulties reconstructing intricate structure. The Chamfer distances $(\times 10^{-3})$ are shown below the synthetic Snowglobe. The Snowglobe is shown in section view.

strategy respectively. This leaves NeUDF [21] the only method that is theoretically capable of rendering multiple layers of surfaces in a single ray.

Figure 7 shows the comparisons with NeUDF [21]. Since the zero distance value of NeUDF would result in opaque surface, the minimum distances of transparent objects learned by NeUDF are also

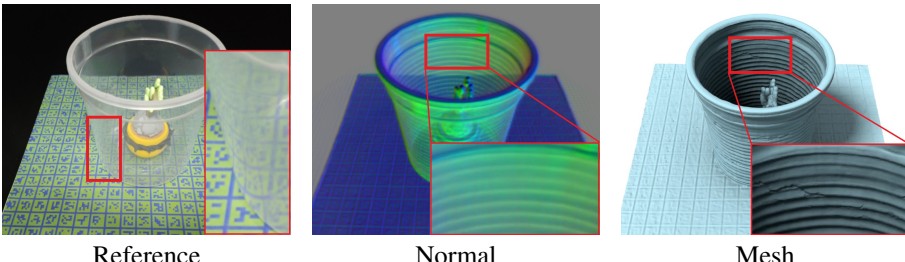

| Reference | Normal | Mesh |

Figure 8: The reflections and refractions in the input multi-view images (left) can lead to ambiguities in distance field learning (middle), preventing our method from extracting the desired surface (right). This phenomenon is particularly pronounced in the reconstruction of cylinders or spheres under complex lighting conditions.

positive. We use DCUDF to extract surface instead of the default MeshUDF [42] used by NeUDF, because MeshUDF could only extract the zero iso-surface. The models reconstructed by NeUDF are over-smoothed and lack many intricate structures. Qualitative measurement of the synthetic model also shows that NeUDF results in a larger Chamfer distance. This is due to the volatile nature of UDF learning, requiring additional regularizers for successful convergence, often sacrificing the reconstruction fidelity.

**Limitations.** Although our method has been effectively validated on both synthetic and real-world data, it cannot handle all use cases. Our method, together with $\alpha$Surf [29], allows for simultaneous reconstruction of opaque and transparent objects. Other works either focus on the reconstruction of opaque objects, or pure-glass objects with refraction and reflection under certain assumptions. However, our method is not designed to handle the cases with complex lighting conditions like heavy refraction or reflection. As shown in Figure 8, when using NeuS [4] as the backbone, scenes with reflection and refraction may yield ambiguous distance fields, preventing the acquisition of the ideal surface. For these situations, on the one hand, improving the lighting conditions to minimize the occurrence of refraction and reflection can be considered. In our experiments, we used polarizer to reduce reflection and model only thin transparent objects that have as little refraction as possible. Meanwhile, it is also possible to use existing reflection removal algorithms [43]. On the other hand, replacing the backbone with models like Ref-NeuS [28] or ReNeuS [8] (which focus on opaque object reconstruction but not the transparent object itself) could be considered. This will be one of our future research directions.

## 5 Conclusion

Overall, $\alpha$-NeuS presents a new perspective on NeuS. We proved the unbiasedness of NeuS for transparent objects and extended the capability of NeuS to transparent surface and opaque surface reconstruction by proposing a unified theoretical and practical framework. Based on DCUDF, we extract the unbiased transparent surface and opaque surface simultaneously for model reconstruction. We established a benchmark consisting of 5 synthetic and 5 real world scenes for validation. Our experiments have demonstrated the effectiveness of our proposed method, and its practical potentials.

## Acknowledgments and Disclosure of Funding

This work was supported in part by the National Key R&D Program of China (2023YFB3002901), in part by the Basic Research Project of ISCAS (ISCAS-JCMS-202303), in part by the Major Research Project of ISCAS (ISCAS-ZD-202401), in part by the Ministry of Education, Singapore, under its Academic Research Fund Grants (MOE-T2EP20220-0005 & RT19/22) and the RIE2020 Industry Alignment Fund–Industry Collaboration Projects (IAF-ICP) Funding Initiative, as well as cash and in-kind contribution from the industry partner(s).

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

# A Proof of Theorem 1

**Theorem 1.** *Assuming a single ray-plane intersection, if the rendered opacity $\alpha \leq 0.5$, the learned distance field reaches a local minimum which is non-negative, and the corresponding color weight maximum aligns with the distance local minimum. Otherwise, the distance local minimum is smaller than zero, and the corresponding color weight maximum aligns with the zero iso-surface.*

*Proof.* Before we start our proof, we first provide a discussion of the rendered opacity $\alpha$. We define the rendered opacity $\alpha$ as the integral of the color weights $w(t)$ along the entire ray $\mathbf{r}(t) = \mathbf{o} + t\mathbf{d}$. This definition coincides with the code that calculates the opacity, found in NeuS's source code. Technically speaking, $\alpha$ can be defined by Eqn. (3).

$$\alpha = \int_0^{+\infty} w(t)\,\mathrm{d}t \tag{3}$$

A discovery made by HF-NeuS [6] connects the color weight $w(t)$ with the accumulated transmittance $T(t)$, as shown in Eqn. (4).

$$\frac{\mathrm{d}}{\mathrm{d}t}T(t) = -T(t)\sigma(\mathbf{r}(t)) = -w(t) \tag{4}$$

Hence, we can simplify the integral of the color weights $w(t)$ to the integral of the derivative of the accumulated transmittance $T(t)$, connecting the rendered opacity $\alpha$ with $T(t)$ itself directly, shown in Eqn. (5).

$$\begin{aligned}
\alpha = \int_0^{+\infty} w(t)\,\mathrm{d}t &= \int_0^{+\infty} -\frac{\mathrm{d}}{\mathrm{d}t}T(t)\,\mathrm{d}t \\
&= T(0) - \lim_{t\to+\infty} T(t) \\
&= 1 - \lim_{t\to+\infty} T(t)
\end{aligned} \tag{5}$$

Our proof is divided into two parts. In the first part, we prove that the distance field we described in the theorem could lead to different rendered opacity $\alpha$. After that, in the second part, we prove the unbiasedness claimed in the theorem.

**Part I** We first provide a recapitulate of the density function defined in NeuS [4]. NeuS [4] and HF-NeuS [6] both derive the same density formula, shown in Eqn. (6).

$$\tilde{\sigma}(\mathbf{r}(t)) = \frac{-\frac{\mathrm{d}}{\mathrm{d}t}\Phi_s(f(\mathbf{r}(t)))}{\Phi_s(f(\mathbf{r}(t)))} = -s\left(1 - \Phi_s(f(\mathbf{r}(t)))\right)\cos(\theta) \tag{6}$$

where $\Phi_s(\cdot)$ is sigmoid function parameted by $s$, and $\theta$ is the angle between the ray direction and the gradient of the distance field $f$, and $\cos(\theta) < 0$. NeuS [4] and HF-NeuS [6] further clip $\tilde{\sigma}$ against 0 wherever $\cos(\theta)$ becomes positive, to ensure non-negative density values, shown in Eqn. (7). This process is effectively equivalent to back-face culling technique used in mesh rasterization pipelines.

$$\sigma(\mathbf{r}(t)) = \max(\tilde{\sigma}(\mathbf{r}(t)), 0) \tag{7}$$

We consider the scenario where the ray intersects with only one front-facing plane at $t = t_0$. Let the local minimum of the distance field be $m$, the signed distance function $f(\mathbf{r}(t))$ can be explicitly written as Eqn. (8). We provide readers with Figure 9 to help understand the different terms in Eqn. (8).

$$f(\mathbf{r}(t)) = \begin{cases} \left|\left|(t - t_0) \cdot |\cos(\theta)|\right|\right| + m, & m \geq 0 \\ \left|\left|\left(t - t_0 - \frac{-m}{|\cos(\theta)|}\right) \cdot |\cos(\theta)|\right|\right| + m, & m < 0 \end{cases} \tag{8}$$

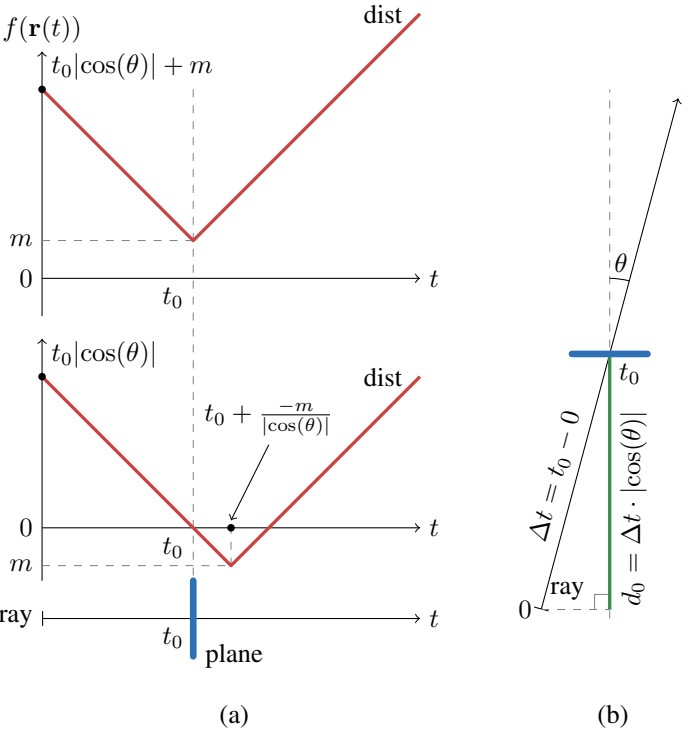

Figure 9: Illustration for Eqn. (8). (a) The two red curves represent the two cases of the distance functions when $m \geq 0$ and $m < 0$ respectively. (b) is an illustration showing the relationship of the distance $d_0$, $t_0$ and $\cos(\theta)$.

When $m \geq 0$, direct calculation of $\alpha$ is shown in Eqn. (9), by substituting the variable $t$ with $sf(\mathbf{r}(t))$.

$$
\begin{aligned}
\alpha &= 1 - \lim_{t \to +\infty} T(t) \\
&= 1 - \exp\left(-\int_0^{+\infty} \sigma(\mathbf{r}(t))\, dt\right) \\
&= 1 - \exp\left[-\int_0^{+\infty} \max\left(\frac{-se^{-sf(\mathbf{r}(t))}}{1 + e^{-sf(\mathbf{r}(t))}} \cdot \cos(\theta), 0\right) dt\right] \\
&= 1 - \exp\left(-\int_0^{t_0} \frac{-se^{-sf(\mathbf{r}(t))}}{1 + e^{-sf(\mathbf{r}(t))}} \cdot \cos(\theta)\, dt\right) \\
&= 1 - \exp\left(-\int_{s(t_0|\cos(\theta)|+m)}^{sm} \frac{-e^{-sf(\mathbf{r}(t))}}{1 + e^{-sf(\mathbf{r}(t))}}\, d(sf(\mathbf{r}(t)))\right) \\
&= 1 - \frac{e^{sm} + e^{-st_0|\cos(\theta)|}}{1 + e^{sm}} \\
&= \frac{1 - e^{-st_0|\cos(\theta)|}}{1 + e^{sm}},
\end{aligned}
\tag{9}
$$

and $m \geq 0$ leads to the resulting range of $\alpha \in \left(0, \frac{1-e^{-st_0|\cos(\theta)|}}{2}\right]$. Note that $t_0|\cos(\theta)|$ is the distance between the camera origin and the surface. If we denote that distance as $d_0$, we can rewrite Eqn. (9) as $\alpha = \frac{1-e^{-sd_0}}{1+e^{sm}} \in \left(0, \frac{1-e^{-sd_0}}{2}\right]$.

It's worth noting that when $m = 0$, the distance field is the corresponding UDF of the scene. 2S-UDF [24] discovers that naively applying NeuS's density function on UDFs will lead to transparency in rendered surfaces. Our theoretical result explains this observation.

When $m < 0$, direct calculation of $\alpha$ is shown in Eqn. (10), calculated in a similar manner.

$$
\begin{aligned}
\alpha &= 1 - \lim_{t \to +\infty} T(t) \\
&= 1 - \exp\left( - \int_0^{+\infty} \sigma(\mathbf{r}(t)) \, \mathrm{d}t \right) \\
&= 1 - \exp\left[ - \int_0^{+\infty} \max\left( \frac{-se^{-sf(\mathbf{r}(t))}}{1 + e^{-sf(\mathbf{r}(t))}} \cdot \cos(\theta), 0 \right) \mathrm{d}t \right] \\
&= 1 - \exp\left( - \int_0^{t_0 + (-m)/|\cos(\theta)|} \frac{-se^{-sf(\mathbf{r}(t))}}{1 + e^{-sf(\mathbf{r}(t))}} \cdot \cos(\theta) \, \mathrm{d}t \right) \\
&= 1 - \exp\left( - \int_{st_0|\cos(\theta)|}^{sm} \frac{-e^{-sf(\mathbf{r}(t))}}{1 + e^{-sf(\mathbf{r}(t))}} \, \mathrm{d}(sf(\mathbf{r}(t))) \right) \\
&= 1 - \frac{1 + e^{-st_0|\cos(\theta)|}}{1 + e^{-sm}},
\end{aligned}
\tag{10}
$$

and $m < 0$ leads to the resulting range of $\alpha \in \left( \frac{1 - e^{-st_0|\cos(\theta)|}}{2}, 1 \right)$. Using the same notation above, we can rewrite Eqn. (10) as $\alpha = 1 - \frac{1 + e^{-sd_0}}{1 + e^{-sm}} \in \left( \frac{1 - e^{-sd_0}}{2}, 1 \right)$.

In this case, each front-facing plane's associated back-facing plane, which is not rendered due to the clipping of the density, begin to separate from the front-facing plane. A larger $m$ means a further-away back face. When the back face is infinitely away, $m$ will approach $-\infty$. The rendered opacity $\alpha$ will then approach 1, which means fully opaque. This scene can be considered a single ray-plane intersection, and is the scene on which NeuS [4] and HF-NeuS [6] base their density function derivation.

Combining the two cases together, different choices of $m$ will cover every rendered opacity $\alpha$ from 0 to 1. The watershed $\alpha$ of the two cases is $\frac{1 - e^{-sd_0}}{2}$. Since after training, the parameter $s$ converges to a large number, and the origin of the ray is usually away from the surface (meaning that $d_0$ is relatively large), the watershed $\alpha$ is approximately 0.5.

**Part II**  Only the places where $\cos(\theta) < 0$ contributes to the rendered color, and the color weight function $w(t)$ is continuous and smooth in this region. Therefore, Eqn. (11) shows the derivative of $w(t)$ with respect to $t$.

$$
\begin{aligned}
w'(t) &= [T(t)\sigma(\mathbf{r}(t))]' \\
&= T'(t)\sigma(\mathbf{r}(t)) + T(t)\sigma'(\mathbf{r}(t)) \\
&= (-T(t)\sigma(\mathbf{r}(t)))\sigma(\mathbf{r}(t)) + T(t)\sigma'(\mathbf{r}(t)) \\
&= -T(t)\sigma^2(\mathbf{r}(t)) + T(t)\sigma'(\mathbf{r}(t)) \\
&= T(t)\left[ \sigma'(\mathbf{r}(t)) - \sigma^2(\mathbf{r}(t)) \right]
\end{aligned}
\tag{11}
$$

Should the local maximum occur at $t = t^*$, we get $w'(t^*) = 0$. Since $T(t) > 0$ is always true, Eqn. (12) should hold true:

$$
\sigma'(\mathbf{r}(t^*)) = \sigma^2(\mathbf{r}(t^*))
\tag{12}
$$

Since

$$
\begin{aligned}
\sigma'(\mathbf{r}(t^*)) &= \left[ -s\left( 1 - \frac{1}{1 + e^{-sf(\mathbf{r}(t^*))}} \right) \cos(\theta) \right]' \\
&= \frac{s^2 \cos^2(\theta) e^{-sf(\mathbf{r}(t^*))}}{\left( 1 + e^{-sf(\mathbf{r}(t^*))} \right)^2},
\end{aligned}
\tag{13}
$$

and

$$\sigma^2(\mathbf{r}(t^*)) = \frac{s^2 \cos^2(\theta) e^{-2sf(\mathbf{r}(t^*))}}{\left(1 + e^{-sf(\mathbf{r}(t^*))}\right)^2}, \tag{14}$$

solving Eqn. (12), we get

$$1 = e^{-sf(\mathbf{r}(t^*))} \implies f(\mathbf{r}(t^*)) = 0 \tag{15}$$

When $m < 0$, the zero distance field value position exists at $t^* = t_0$, and the maximum weight position aligns with the zero distance field value position.

When $m = 0$, the position of the distance field minimum value is also the position of the zero distance value. Therefore, the maximum weight occurs at the zero distance position, which is also the distance field minimum position.

When $m > 0$, the distance field is nowhere zero. In this scenario, we have $\forall t < t_0$,

$$m > 0$$
$$\implies \sigma^2(\mathbf{r}(t)) = \sigma'(\mathbf{r}(t)) \cdot e^{-sf(\mathbf{r}(t^*))} < \sigma'(\mathbf{r}(t)) \cdot e^{-s \cdot 0} = \sigma'(\mathbf{r}(t)) \tag{16}$$
$$\implies w'(t) > 0.$$

This means that the color weight is continuously increasing until the distance field reaches its minimum, implying that the maximum position of the color weight is aligned with the position of the distance field minimum.

And that completes the proof. □

## B  Additional Experimental Results

### B.1  More comparisons with NeUDF [21]

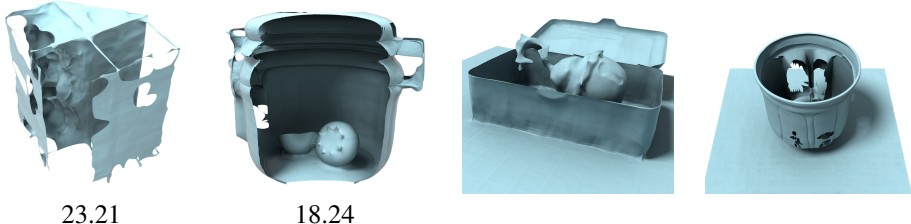

23.21                18.24

Figure 10: More results of NeUDF [21]. The Chamfer distances ($\times 10^{-3}$) are shown below synthetic models. The Jar is shown in section view.

We show more results on NeUDF [21] whose surfaces are extracted by DCUDF [11] in Figure 10. Generally speaking, compared with the results of our $\alpha$-NeuS in the paper, the NeuS backbone outperforms the NeUDF backbone in quantitative and qualitative measures.

### B.2  Empty transparent object

The majority of our data focuses on the objects where transparent and opaque share roughly the same proportion. We include a synthetic case where the vast majority of the object is transparent and without refraction or reflection. The results are shown in Figure 11.

### B.3  Experimental results on DTU dataset [44]

We also conduct experiments on DTU dataset [44], which is an opaque object dataset. Theoretically, our method when applied to pure opaque dataset, is essentially equivalent to the original NeuS [4]. We present the results on selected data in Figure 12. Although the SDF learned by NeuS is not completely clean and may oscillate inside the object, it does not interfere with the mesh extraction process with DCUDF [11], as shown in the top row of Figure 12. Moreover, DCUDF can resolve

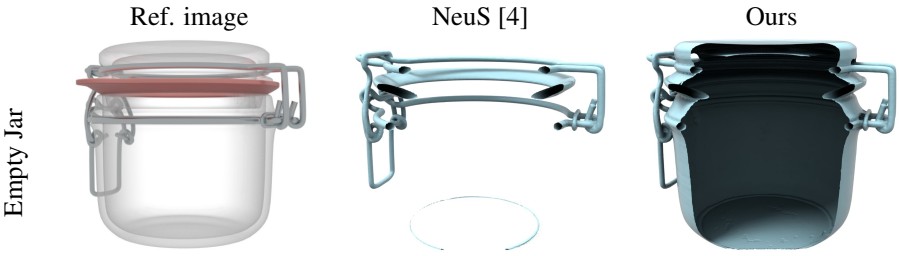

Figure 11: An empty transparent jar (section view).

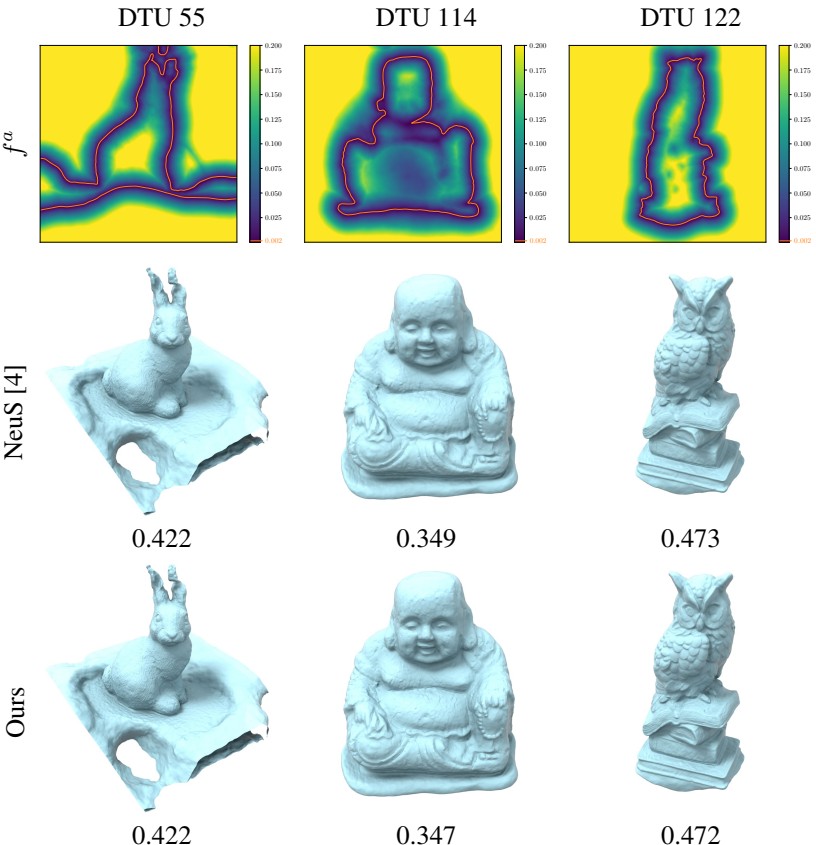

Figure 12: Quantitative comparison on selected DTU dataset [44]. The Chamfer distances ($\times 10^{-3}$) are shown below each model. The slices of the absolute values of the SDF are shown and truncated to 0.2 for better visualization. The orange line is the $r = 0.002$ level set used in DCUDF.

mesh vertices more accurately than marching cubes [10], which relies on interpolation to approximate the zero-value location between grid vertices, therefore there are cases where the mesh extracted by DCUDF can achieve a lower Chamfer distance than those by marching cubes. Similar results are also reported by the authors of DCUDF [11] (Table 3).

## C    Acknowledgments of Dataset

We thank the vibrant Blender community for heterogeneous models that drive our experiments. Specifically, we thank Rina Bothma for creating the bottle model, Joachim Bornemann for creating the case model, Rudolf Wohland for creating the jar model, and Eleanie for creating the strawberry model which is put inside the jar. These models are released under a royalty-free license described

at `https://www.blenderkit.com/docs/licenses/`. Furthermore, we thank MrSorbias for a beginner-friendly tutorial on creating a snow globe in Blender on YouTube.

