# OpenReview forum: "From Transparent to Opaque: Rethinking Neural Implicit Surfaces with $\alpha$-NeuS"
_NeurIPS.cc/2024/Conference — NeurIPS 2024 poster_

### Official Review · Reviewer_GRsP · 2024-07-03

**Soundness:** 2
**Presentation:** 1
**Contribution:** 2
**Rating:** 2
**Confidence:** 3

**Summary:**

The paper proposes an extension of NeuS to also support transparent surfaces. It states a proposition that claims that NeuS is also unbiased for transparent surfaces, proposes an approach for simultaneous extraction of opaque and transparent surfaces, and presents a benchmark dataset.

**Strengths:**

(1) The results are good. The proposed approach successfully reconstructs the opaque and transparent surfaces simultaneously.

(2) The paper presents a benchmark dataset, which may help the empirical standardization of the field and paper writing in consequence.

**Weaknesses:**

(1) The current paper organization and dependency severely hinders its appreciation. It should have at least a background / overview section with a minimum of presentation and intuition behind the core concepts. I understand that this paper is an extension of NeuS, but assuming that reading NeuS is an unavoidable prerequisite to conceive an overview of the method is not a gentle choice regarding readers or reviewers. For example, the Abstract, Introduction, and Method sections refer to the concept of unbiased surfaces, which is only introduced at the second paragraph of section 3. Moreover, that is a loose definition since it depends on the concept of rendering weights, which is not introduced at all. The expression rendering weights is ambiguous since it could means a plethora of concepts in the context of the paper. Presenting the equation beforehand and pointing to $\omega(t)$ would remove the ambiguity.

(2) Also in the ambiguity realm, Theorem 1 is not a theorem per se in its current presentation. To be a theorem, a proposition should be a clearly stated assertion that is precise and unambiguous. To be precise, all symbols must be precisely defined beforehand or in the theorem body. Theorem 1 is based on the weight function $\omega(t)$ presented in NeuS, but its current assertion is plain English, without any precise definitions provided. The proposition is not currently verifiable in that form. Overall the paper has a significant problem stating its definitions and symbols.

(1) and (2) are the core reasons behind my current rating and, unfortunately, I do not think they could be solved without substantial rewriting and in rebuttal time. I think the problem addressed is important, and also that the paper has good results, but it is based on a core theorem that is not correctly stated. I advise the authors to rewrite the paper, following the guidelines:

* Please take a more didactic approach. In practice, at least a background / overview section should be present. The paper should be self-contained in some extent. That would broaden the public for the paper, since readers could understand the paper in different levels and in increasingly difficulty. It would also ease the review process, possibly earning more favor in future reviews.

* Please precisely present the symbols and equations before the theorem statement to remove ambiguities. The usual form of a theorem is "Let {definitions and symbols}. {conditions using the symbols} then {conclusions using the symbols}." In case the logic is bidirectional, the form is "Let {definitions and symbols}. {conditions / conclusions using the symbols} if and only if {conditions / conclusions using the symbols}". Following those standards helps to smooth the reading flow and helps the reader to verify whether the theorem holds. Since the theorem is a core component of the paper, including a mathematician author or making one reviewing the paper before the submission could also be a good idea.

* Another approach would be to remove the theorem and make a discussion based on empirical evidence. The current results are good, arguably sufficient for acceptance if the paper is written in such standpoint.

(3) The paper criticizes $\alpha$Surf in a few locations, saying it introduces noise and artifacts. I can see such phenomena in that reference, however I do not think it is good practice to criticize another work without showing comparison experiments. Since the work is still unpublished and the code source is unavailable I would like to politely ask the authors to remove such critics. A more polite way to contextualize would be to say that $\alpha$Surf is a concurrent work, without judging its results.

**Questions:**

(1) I would like to see a quantitative evaluation against NeUDF too. Even though NeUDF seems to smooth the surface, $\alpha$-NeuS also seems to introduce noise. I am curious about which one is closer to the ground truth surface.

**Limitations:**

Yes, the paper has a paragraph about limitations.

---

> ### Author Rebuttal · Authors · 2024-08-07
>
> We thank the reviewer for the comments.
>
> **Q: Overview of NeuS**
>
> A: We appreciate your feedback and will add the overview of NeuS in our revision. We prepare an initial overview as follows:
>
> NeuS is a surface reconstruction method based on Neural Radiance Field (NeRF) that uses volume rendering technique, which is differentiable, to render the image and optimize the scene representation. In the realm of volume rendering, the color of individual pixels is determined by the color of individual rays that shoots from the camera origin $\mathbf{o}$ and passes through the corresponding pixel on the imaginary image plane in the 3D space, parametered by $\mathbf{r}(t)=\mathbf{o} + t\mathbf{d}$. The color of the ray, $C(\mathbf{r})$, is determined by a weighted integral of colors $c(\mathbf{r}(t), \mathbf{d})$ along every point of the ray,
> $$C(\mathbf{r}) = \int_0^{+\infty}w(t)c(\mathbf{r}(t), \mathbf{d}) \mathrm{d}t.$$
> Note that in practice, the upper bound of integral is not infinity but rather a fixed big value. This makes sense because it is expected that there is nothing of interest far away from the camera.
>
> In the volume rendering framework, the assigned color weights $w(t)$ of each point on the ray is determined by the volume density $\sigma(t)$ of the point and the accumulated transmittance $T(t)$ at this point,
> $$w(t) = T(t)\sigma(\mathbf{r}(t)).$$
>
> The accumulated transmittance $T(t)$ at point $\mathbf{r}(t)$ can be construed as the possibility that a ray, starting from its origin (camera position $\mathbf{o}$) and traversing along the ray path, could arrive at this point without being hindered somewhere before. $T(t)$ is formulated as
> $$T(t) = \exp\left\lparen-\int_0^t\sigma(\mathbf{r}(u))\mathrm{d}u\right\rparen.$$
>
> The color $c(\mathbf{r}(t), \mathbf{d})$ is predicted by a neural network. But in contrast to NeRF where the neural network directly predict the volume density $\sigma(\mathbf{r}(t))$, NeuS predicts the signed distance value at the point $f(\mathbf{r}(t))$ instead. NeuS further leverages a mapping to map the signed distance $f(\mathbf{r}(t))$ to the volume density $\sigma(\mathbf{r}(t))$ at this point. The tight mapping of the signed distance field and the volume density distribution enables high fidelity SDF learning through back propagation.
>
> When tying the object geometry with volume rendering, NeuS puts forward two major requirements for the color weight function $w(t)$: unbiased and occlusion-aware. For the rendering to be "unbiased", the color weight function $w(t)$ should attain locally maximal value at the surface intersection point $\mathbf{r}(t_0)$, which, in the opaque scenario, corresponds to the zero level set of the SDF, i.e., $f(\mathbf{r}(t_0)) = 0$. NeuS proposes a mapping from distance to density that is unbiased in the first order. For the rendering to be occlusion-aware, whenever $0 < t_1 < t_2$ and $f(\mathbf{r}(t_1)) = f(\mathbf{r}(t_2))$, $w(t_1)$ should be greater than $w(t_2)$. This aligns with the everyday experience that the object nearer to the camera ($t_1$) should occlude the one ($t_2$) behind it. An overview of the distance-to-density mapping function proposed by NeuS can be found in Appendix A (Eqn. (6) and Eqn. (7)).
>
> We use $\alpha$ to denote the rendered opacity. The $\alpha$ of a pixel is defined as the integral of all color weights $w(t)$ along the corresponding ray,
> $$\alpha = \int_0^{+\infty} w(t) \mathrm{d}t.$$
>
> **Q: The presentation of Theorem 1.**
>
> A: We appreciate your feedback and revise Theorem 1 as follows:
>
> Theorem 1: Consider a ray $\mathbf{r}(t)$ intersecting a plane at a point $\mathbf{r}(t_0)$. Let $f(\mathbf{r}(t))$ be the learned distance function and $w(t)$ the rendering color weight. If $f(\mathbf{r}(t))$ attains its minimum at $t=t_m$ and $\alpha$ is the plane's opacity:
> + If $\alpha<0.5$, then $t_0=t_m$, $f(\mathbf{r}(t_m))>0$, and $w(t)$ peaks at $t_m$.
> + If $\alpha \geq 0.5$, then $f(\mathbf{r}(t_m)) \leq 0$ and $f(\mathbf{r}(t_0))=0$, positioning $w(t)$'s maximum at $t$ with $f(\mathbf{r}(t))=0$.
>
> **Q: The quantitative evaluation with NeUDF.**
>
> A: Please refer to the combined responses to all reviewers for detailed evaluations.
>
> **Q: A more polite way to contextualize would be to say that  $\alpha$-Surf is a concurrent work, without judging its results.**
>
> A: Thank you for the guidance. We will revise our discussion to reference $\alpha$Surf as a concurrent work. As mentioned by reviewer ruwJ, $\alpha$Surf has recently released a preliminary version of its code, which we have evaluated and discussed in Figure 1 of the supplementary PDF. We observed that $\alpha$Surf outperforms NeuS and NeUDF+DCUDF on the synthetic dataset, judged by the average Chamfer distances. We hypothesize that the observed surface noise in $\alpha$Surf's results arises from the use of alpha shapes [1] to generate meshes from point clouds. Given the extensive research on producing more reliable meshes from point clouds [2][3][4][5], we anticipate future improvements in $\alpha$Surf results. Given that the currently released code of $\alpha$Surf is preliminary, we regard our existing comparison with $\alpha$Surf as initial findings.
>
> [1] Edelsbrunner et al. On the shape of a set of points in the plane, IEEE Transactions on Information Theory, 29 (4): 551–559, 1983
>
> [2] Zhou et al. Learning a more continuous zero level set in unsigned distance fields through level set projection. ICCV. 2023: 3181-3192.
>
> [3] Liu et al. Ghost on the Shell: An Expressive Representation of General 3D Shapes. ICLR, 2024.
>
> [4] Chen et al. Neural dual contouring. ACM Transactions on Graphics (TOG), 2022, 41(4): 1-13.
>
> [5] Wang et al. Hsdf: Hybrid sign and distance field for modeling surfaces with arbitrary topologies. NeurIPS, 2022, 35: 32172-32185.

---

> > ### Comment · Reviewer_GRsP · 2024-08-07
> >
> > Thank you for considering the review and for the responses.

---

> > > ### Author Response · Authors · 2024-08-08
> > >
> > > Dear Reviewer GRsP,
> > >
> > > Thank you for dedicating time to review our manuscript. We appreciate your feedback regarding the overview of NeuS and the symbolic expression of Theorem 1. We have carefully addressed these points in our rebuttal and believe that the necessary revisions are straightforward.
> > >
> > > We noted that subsequent to our rebuttal submission, the evaluation of our paper shifted from 'Reject' to 'Strong Reject.' Could you please provide further clarification or specific reasons for this change in the assessment? We are eager to understand your concerns more fully and to address them appropriately in our revisions.
> > >
> > > Thank you.
> > >
> > > The authors

---

> ### Comment · Reviewer_GRsP · 2024-08-08
>
> The main claims of the paper are theoretical, the entire Method section is based on Theorem 1. When the authors organize the paper that way, they should be aware of the burden of rigorous Math checking in the review. As I said before, Theorem 1 was not stated as a theorem in the original manuscript, making it impossible to check. It was highly ambiguous, with several core definitions missing. My original rating was Strong Reject because that is a major technical and methodological flaw, turning it impossible to adequately evaluate the paper. I turned it a Reject because I had hope the other reviews and rebuttal could point out other perspectives that could make me reevaluate the rating. After carefully reading them and reflecting, that was not the case.
>
> I appreciate that the authors provided a new statement for the theorem, however it is not possible for me to rigorously do that checking in the rebuttal discussion time frame because I have other five reviews in NeurIPS, and submissions. Unfortunately, there was nothing the authors could have done to increase the rating, given the situation. I believe it is risky for NeurIPS to accept a theoretical paper without properly checking the core theorem.
>
> I maintain my advice that the authors should create a new version based on a rigorously stated and proven theorem with adequate context or change the paper structure to not focus on theoretical claims. For example, the original NeuS has a different structure, with the theorem not being protagonist in the claims. Even though I believe every theorem should be rigorously checked (even the ones in supplementary), there is a sad tendency in Machine Learning papers to not treat theorems with proper attention. In some extent, I consider that a disrespect to the field of Mathematics and to mathematicians. Unfortunately, the manuscript crossed the boundary of what is acceptable in a theoretical standpoint.

---

> ### Author Response · Authors · 2024-08-09
>
> The mathematically rigorous proof of the theorem is in the technical appendices of our original submitted paper. Since NeuS is a well-established work in the field, we assume familiarity among our readership. We believe that the fewer symbols the easier it is to understand, so the theorem was originally presented in plain language in our original paper. As requested by the Reviewer GRsP, we revise the theorem symbolically, and present it in a mathematically rigorous way. Here we also outline the proof briefly here for clarity:
>
> (1) Opacity-Distance Relationship: Utilizing the volume rendering equation and NeuS's distance to density function, we establish a correlation between opacity and learned distance. Specifically, we demonstrate that opacity $\leq 0.5$ when the learned distance reaches a non-negative local minimum, and opacity $> 0.5$ when the local minimum is negative.
>
> (2) Color Weight Maximum: We analyze the derivatives of the color rendering weight relative to the ray parameter $t$. We establish that the maximum color weight coincides with the local minimum distance when it is non-negative, and with the zero iso-surface when the local minimum distance is negative.
>
> Reviewer GRsP raised concerns about what he/she described as "ambiguous" concepts, including rendering weights and unbiasedness. Notice that the rendering weight $w(t)$ is fundamental to the volume rendering equations used in prominent models such as NeRF and NeuS. Similarly, "unbiasedness" is a pivotal concept in NeuS, now considered standard in neural rendering of implicit surfaces. Since these terms are well-established within the field, we chose not to redefine them in our initial submission. Recognizing that these concepts might be challenging for readers who are not familiar with NeuS, we plan to include a detailed overview of NeuS in the revised manuscript, as indicated in our rebuttal.
>
> We respectfully disagree with Reviewer GRsP on the notion that machine learning papers neglect rigorous theorem treatment. Contrary to this view, all other reviewers have acknowledged the importance of our theoretical contribution. As pointed by Reviewer 6LF5, our paper "provides a rigorous theoretical analysis, extending the unbiasedness proof of NeuS to cover a range of material opacities from fully transparent to fully opaque. This theoretical work completes and expands the framework of NeuS."
>
> Beyond theoretical contributions, our work includes practical advances. We have developed an algorithm that outperforms existing methods and the concurrent $\alpha$Surf approach. We also introduce a real-world dataset featuring semi-transparent objects, showcasing the practical effectiveness of our approach on these realistic scenes.

---

### Official Review · Reviewer_vpPg · 2024-07-05

**Soundness:** 3
**Presentation:** 2
**Contribution:** 3
**Rating:** 6
**Confidence:** 3

**Summary:**

This paper proposes an adaptation to NeuS to extract semi-transparent surfaces. The method relies on training the original NeuS but with a modification to the mesh extraction. Instead of using marching cube; this paper proposes to extract the 0 level of the SDF field and also its local extrema that represent semi-transparent surfaces.

The contributions are:
- A theoretical derivation to show that the proposed method is sound
- A new method to extract mesh

The method is evaluated on a custom dataset with both synthetic and real scenes of semi transparent objects.

**Strengths:**

- The paper is interesting, it clearly improves over vanilla NeuS.
- The paper introduces a new dataset with transparent objects, the author say this dataset will be released on github

**Weaknesses:**

- Experiments and data:
  - It would be nice to have an additional experiment showing the geometric accuracy of the new mesh extraction procedure with usual data, like DTU. I expect this method to perform worse than NeuS on DTU since this method can extract many spurious local minima of the SDF that are ignored with Marching Cube.
  - While the proposed dataset is interesting, it may be too easy compared to real world. The synthetic data looks like specularities are disabled. However specularities are most present on transparent objects (glass, water) and this may be a harder challenge to solve. As stated in the limitation section, the proposed method cannot deal with reflections.



- Presentation
  - There could be a discussion about the role of the temperature parameter $s$ in equation (1). This parameter is used to map a SDF value to an opacity. It is introduced in Neus as a method to have a smooth optimization, starting from fuzzy surfaces and finishing with sharp ones. However in this paper s is also used to make sure nonzero SDF still produce nonzero opacity.
  - From my understanding, this method follows NeuS training but this is never clearly stated
  - What are the normal maps in Fig 4, 6, it looks like normals from the proposed method but it is not written in the caption.
  - Lack of explanations about the optimization (Section 3.2). The paper shows two objective functions (l 167 and 176) it is not clear which one is used. Also there are no explanation about the optimization itself, the solver, stopping criterion, etc.

**Questions:**

See limitations: the paper could benefit from an improved presentations.

I expect from the rebuttal:
- Results and discussion on opaque scene (DTU)
- More explanations about the optimization of section 3.2

**Limitations:**

The limiations are correctly stated

---

> ### Author Rebuttal · Authors · 2024-08-07
>
> We thank the reviewer for the comments.
>
> **Q: Results and discussion on opaque models (DTU).**
>
> A: We have added additional results on the DTU dataset in Figure 4 of the supplementary PDF. The results of our $\alpha$-Neus and the original NeuS are similar.
>
> We show the absolute values of the SDF learned by NeuS, in which DCUDF performs optimization. We observe some local minima/maxima within the model in the distance field. Most of these local minima/maxima appear to be above $r$ and far away from the surface. Local minima larger than $r$ don't affect DCUDF. Thus the influence of local minima is small. In addition, DCUDF can resolve mesh vertices more accurately than Marching Cubes at the same resolution. This has been demonstrated by DCUDF -- see Table 3 of DCUDF paper -- resulting in better Chamfer distance values in some cases.
>
> **Q: More explanations about the optimization in Section 3.2.**
>
> A: Please refer to the combined responses to all reviewers for detailed evaluations.
>
> **Q: The normal maps in Figures 4 and 6.**
>
> A: The normal maps displayed in Figures 4 and 6 were generated using the official NeuS codebase for calculating normal maps. We will specify this clearly in the revision.
>
> **Q: The setting of NeuS.**
>
> A: Please refer to the combined responses to all reviewers for detailed evaluations.
>
> **Q: The role of $s$ in Equation (1).**
>
> A: In NeuS, $s$ is a learnable parameter that gradually converges to a large value over the course of training. A larger $s$ value sharpens the edges and faces in the reconstructed model, enhancing overall quality. However, in practice, $s$ cannot increase indefinitely due to numerical computation constraints, such as the number of sample points. This caps $s$ at relatively high but finite values, which allows non-zero distance values to influence color calculation along the ray. Colors are derived from the the weighted sum of sampled radiance at points along the rays, and different distance minimum values contributes to achieving different levels of opacity.
>
> We believe that the tendency of a larger $s$, combined with the color loss and the actual situations with MLP and numerical calculation will achieve a balance that leads to the best results.

---

### Official Review · Reviewer_6LF5 · 2024-07-11

**Soundness:** 4
**Presentation:** 4
**Contribution:** 4
**Rating:** 8
**Confidence:** 4

**Summary:**

The paper proposed a extended version of NeuS that can reconstruct both transparent and opaque surfaces at the same time with unbiasedness. The unbiasedness is guaranteed by theoretical proof and a companying surface extraction method is also proposed. The authors evaluated their method on a small benchmark and demonstrated the methods’ effectiveness and superiority over other methods.

**Strengths:**

1. Solid theoretical foundation: The paper provides a rigorous theoretical analysis, extending the unbiasedness proof of NeuS to cover a range of material opacities from fully transparent to fully opaque. This theoretical work completes and expands the framework of NeuS.
2. Effective surface extraction method: The proposed surface extraction technique, which combines aspects of DCUDF with a novel approach to handling mixed SDF and UDF fields, appears well-designed and effective. It successfully extracts both transparent and opaque surfaces from the learned distance field.
3. Comprehensive evaluation: The paper presents thorough comparative and evaluation results, both qualitative and quantitative. The experiments cover both synthetic and real-world datasets, providing strong evidence for the effectiveness of the proposed method. The quantitative metrics, particularly the Chamfer Distance comparisons, offer clear and convincing support for the method's superiority over baseline approaches.
4. Discussion of limitations: The authors include a discussion of failure cases. The identified limitations, particularly regarding complex lighting conditions with heavy refraction or reflection, are reasonable and understandable given the scope of the work. This discussion provides valuable insights for future research directions.

In summary, this submission presents a solid contribution to the field, combining theoretical advancements, practical algorithmic development, and thorough empirical validation.

**Weaknesses:**

1. The evaluation set is relatively small, adding more synthetic cases would be fine.

**Questions:**

I don’t have any question for this submission.

**Limitations:**

1. Is it possible to include participating media in your pipeline? e.g. if there’s some smog inside a jar, will the unbiasedness still apply? Or can the pipeline be generalized into a stochastic geometry perspective?
2. If possible, can GT of real-world test cases be constructed with some ‘destroy’ to the object, like what NeRO did?

---

> ### Author Rebuttal · Authors · 2024-08-07
>
> We thank the reviewer for the comments.
>
> **Q: The evaluation set is relatively small, adding more synthetic cases would be fine.**
>
> A: Thank you for your suggestion. We added a synthetic example in Figure 3 of the supplementary one-page PDF file as a test of an empty transparent object without refraction and reflection.
>
> **Q: Is it possible to include participating media in your pipeline? e.g. if there’s some smog inside a jar, will the unbiasedness still apply? Or can the pipeline be generalized into a stochastic geometry perspective?**
>
> A: Including semi-transparent media such as smog within a jar could cause the learned distances to oscillate near zero, simulating the presence of smog. The oscillation might result in multiple local minima, complicating surface extraction. If these local minima could be consistently excluded from the DCUDF (e.g., if the minima are all greater than the user-specified iso-value $r$), the intended surface could potentially be extracted. Theoretically, it is feasible to reconstruct the jar by minimizing smog interference, though practically, this approach may not be robust.
>
> **Q: If possible, can GT of real-world test cases be constructed with some "destroy" to the object, like what NeRO did?**
>
> A: Thank you for this insightful suggestion. However, capturing  transparent objects with an RGBD camera poses challenges due to their optical properties. We will explore this possibility in future research endeavors.

---

### Official Review · Reviewer_ruwJ · 2024-07-12

**Soundness:** 3
**Presentation:** 3
**Contribution:** 3
**Rating:** 6
**Confidence:** 3

**Summary:**

The paper presents a theoretically justified method for surface reconstruction of transparent and opaque objects. The authors prove the theorem that the NeuS density function is unbiased even for transparent surfaces. They propose a method based on DCUDF for extracting unbiased surfaces. The method is validated on synthetic Blender datasets and real datasets captured by the authors. The method is compared quantitatively with NeuS at different threshold levels and qualitatively with NeUDF.

**Strengths:**

1. Theorem 1 is an important fact for the 3D reconstruction community as NeuS is a fundamentally justified method for surface reconstruction.
2. The authors presented additional datasets with transparent and opaque objects. The number of open datasets with such properties is limited.
3. The proposed algorithm for NeuS modification is a solid technical contribution that combines NeuS and DCUDF methods.

**Weaknesses:**

1. There are no comparisons of the method on publicly available data with transparent and opaque objects, see Question 2.
2. My concern is that it is not clear whether the method can reconstruct scenes with only transparent objects. Comparison with transparent object methods is limited, see Question 1. The method hasn't been tested on transparent object datasets that have been used previously in the literature, see Question 2.
3. Another concern is the number of competitors. In fact, there is only one quantitative evaluation with NeuS. A quantitative comparison of the method with NeUDF is not presented.

**Questions:**

Questions and suggestions:
1. Related work on the method of 3D reconstruction of transparent objects is limited and should be reviewed. I suggest that the authors review the following papers: "NEMTO: Neural Environment Matting for Novel View and Relighting Synthesis of Transparent Objects", ICCV 2023; "Ray Deformation Networks for Novel View Synthesis of Refractive Objects", WACV-2024; "Volumetric Rendering with Baked Quadrature Fields", 2023. All of these papers have applications to transparent object reconstruction and I would like to see how they relate to considered setup.
2. While some of the previously mentioned papers do not consider datasets with opaque objects, your method is expected to be applicable to datasets with only transparent objects, so I suggest the authors experiment with this setup on publicly available datasets from "Through the Looking Glass": Neural 3D Reconstruction of Transparent Shapes" and NEMTO. It should also be noted that some works consider publicly available data with both transparent and opaque objects, see Fig. 7 and Fig. 9 in the paper "Volumetric Rendering with Baked Quadrature Fields" (https://arxiv.org/pdf/2312.02202v1) from the NeRF-Synthetic and Mip-NeRF-360 datasets respectively.
3. It looks like the authors of "$\alpha$Surf" have released their code (https://github.com/ChikaYan/alphasurf), so I suggest the authors to check it again.
4. It would be better to have an illustration for Eq. 8.
5. Can you clarify training hyperparameters of your method like lambda coefficients, etc and explain their choice?

Typos:
1. Line 44, .e.g -> e.g.

**Limitations:**

The limitations of the method are adequately addressed in the Limitations section.

---

> ### Author Rebuttal · Authors · 2024-08-07
>
> We thank the reviewer for the comments.
>
> **Q: Related work on transparent objects.**
>
> A: Thank you for highlighting several pertinent studies that we had initially overlooked. We will incorporate a discussion of these works in our revised manuscript. The works [1,2] address the reconstruction of fully transparent objects by separating rendering from geometry or considering the variations in rays due to refraction and reflection. Sharma et al. [3] introduces multiple intersections during the rendering process through an additional network. It primarily focuses on efficient rendering rather than geometric reconstruction.
>
> References:
>
> [1] Wang D, Zhang T, Süsstrunk S. Nemto: Neural environment matting for novel view and relighting synthesis of transparent objects. in Proceedings of the IEEE/CVF International Conference on Computer Vision (ICCV). 2023: 317-327.
>
> [2] Deng W, Campbell D, Sun C, et al. Ray Deformation Networks for Novel View Synthesis of Refractive Objects. in Proceedings of the IEEE/CVF Winter Conference on Applications of Computer Vision (WACV). 2024: 3118-3128.
>
> [3] Sharma G, Rebain D, Yi K M, et al. Volumetric Rendering with Baked Quadrature Fields. ECCV, 2024.
>
>
> **Q: Testing datasets with only transparent objects.**
>
> A: As mentioned in the limitations section, we "model only thin transparent objects that have as little refraction as possible". As an extension of NeuS, we primarily discuss NeuS's capability in reconstructing transparent objects. However, since refraction is not considered in NeuS, it is infeasible to use NeuS directly to reconstruct thick transparent objects with strong refraction and reflection. The related works, e.g., NEMTO, reconstruct thick transparent objects with significant refraction and reflection, whose goals are different from ours. We are glad to explore using works like NEMTO as a new backbone to extend the versatility of our method in future work.
>
> Meanwhile, we attach a result of a synthetic empty transparent object without refraction and reflection in Figure 3 of the supplementary one-page PDF file.
>
> **Q: Checking the code of $\alpha$Surf.**
>
> A: Thank you for the reminder. As of our submission date, $\alpha$Surf had not yet released its source code. We are keen to perform experimental comparisons with $\alpha$Surf and have already conducted some preliminary experiments. The results are detailed in Figure 1 of the supplementary PDF file provided.
>
> We notice that the $\alpha$Surf repository is currently marked as "a preliminary repo for submission of supplementary material", indicating that it may not contain the final release code. Consequently, the results from these experiments should be viewed as preliminary.
>
>
> **Q: Illustration for Equation 8.**
>
> A: We added the illustration for Equation (8) to enhance clarity and understanding. Please refer to Figure 5 in the supplementary PDF file. (a) The two red curves represent the two cases of the distance functions when $m\geq 0$ and $m<0$ respectively. (b) is an illustration showing the relationship of the distance $d_0$, $t_0$ and $\cos(\theta)$.
>
> **Q: Clarification of training hyperparameters.**
>
> A: Please refer to the combined responses to all reviewers for detailed evaluations.
>
> **Q: The quantitative evaluation with NeUDF.**
>
> A: Please refer to the combined responses to all reviewers for detailed evaluations.

---

### Author Rebuttal · Authors · 2024-08-07

We would like to thank all reviewers for their insightful and constructive comments.

**Q: The quantitative evaluation with NeUDF.**

A: We have included further experimental results related to NeUDF in Figure 1 and 2 of the supplementary one-page PDF file. Generally, the NeuS backbone outperforms the NeUDF backbone. Moving forward, we plan to explore additional methods within the NeuS family as potential backbones for further testing.

**Q: Training hyperparameters and more explanations about the optimization in Section 3.2.**

A: We followed the recommended configuration for the synthetic dataset by the authors of NeuS, without changing the loss functions or their respective weights.

We conducted our experiments using almost the same setting as DCUDF. DCUDF employs a two-stage optimization process. Initially, the optimization follows the formula mentioned in line 167 for the first stage, and subsequently, the formula in line 176 for the second stage. We will clarify these configurations in the revision of the paper. We performed 300 epochs for step 1 and 100 epochs for step 2 respectively, which is the default setting of DCUDF. We used the VectorAdam [1] optimizer as suggested by DCUDF. We set the weight $\lambda_1=500$ which is different from DCUDF's default setting.  We set $\lambda_2=0.5$ as suggested by DCUDF. We do not use the mini-cut postprocessing in DCUDF.

[1] Ling S Z, Sharp N, Jacobson A. Vectoradam for rotation equivariant geometry optimization. Advances in Neural Information Processing Systems, 2022, 35: 4111-4122.

---

### Author Response · Authors · 2024-08-14
**Thank you for your reviews**

Dear Reviewers,

We would like to express our gratitude for your efforts in reviewing our work and providing us with invaluable feedback. We would like to draw attention to the contributions made by our work.

The reconstruction of thin, transparent objects presents a significant challenge. Our objective is to develop a unified framework for reconstructing thin transparent and opaque objects simultaneously. Our key contributions are as follows:

(1) We generalize NeuS to reconstruct transparent and semi-transparent objects and demonstrate, through a mathematically rigorous proof, that unbiased surfaces lie on either the local minimum distance surface (if the local minimum distance is non-negative) or the zero level set.

(2) We propose a unified computational framework to reconstruct 3D objects from fully transparent to fully opaque.

(3) We construct a benchmark that includes both synthetic and real-world models for evaluation, demonstrating that our method outperforms existing methods.

We thank you again for your invaluable feedback.

Sincerely,
The Authors

---

### Decision · Program_Chairs · 2024-09-25

**Decision:**

Accept (poster)

**Comment:**

This paper was reviewed by four experts in the field.  The paper received the following recommendations: Strong Accept, Weak Accept, Weak Accept, and Strong Reject.  All the reviewers agreed that the results are promising and the new dataset will contribute to the community.  On the theoretical contribution, however, the reviewers could not reach a consensus unfortunately.  The three positive reviewers highlight the theoretical contribution in their reviews, while the other reviewer strongly disagrees. Having read the submission, the reviews, and the author feedback, the AC decided to accept the paper.  Although Theorem 1 might have room to improve its presentation, it is acceptable to present it in a plain English in the main text with providing a detailed proof in the appendix as the authors did in the original submission. In addition to this point, the reviewers did raise some valuable concerns that should be addressed in the final camera-ready version of the paper. The authors are encouraged to make the necessary changes to the best of their ability.